# Bigger Data Approach to Analysis of Essential Oils and Their Antifungal Activity against *Aspergillus niger*, *Candida albicans*, and *Cryptococcus neoformans*

**DOI:** 10.3390/molecules24162868

**Published:** 2019-08-07

**Authors:** Chelsea N. Powers, Prabodh Satyal, John A. Mayo, Hana McFeeters, Robert L. McFeeters

**Affiliations:** 1Department of Chemistry, University of Alabama in Huntsville, Huntsville, AL 35899, USA; 2Aromatic Plant Research Center, 230 N 1200E, Lehi, UT 84043, USA

**Keywords:** *Aspergillus niger*, *Candida albicans*, *Cryptococcus neoformans*, essential oil, antifungal, minimum inhibitory concentration, natural product, exclusionary principle, phytochemicals, bioactivity, drug discovery

## Abstract

With increasing drug resistance and the poor state of current antifungals, the need for new antifungals is urgent and growing. Therefore, we tested a variety of essential oils for antifungal activity. We report the minimum inhibitory concentrations (MIC) values for a common set of 82 essential oils against *Aspergillus niger*, *Candida albicans*, and *Cryptococcus neoformans*. Generally, narrow-spectrum activity was found. However, *C. neoformans* was much more susceptible to inhibition by essential oils with over one-third of those tested having MIC values below 160 ppm. GC-MS analysis showed the essential oils to be chemically diverse, yet, the potentially active major constituents typically fell into a few general categories (i.e., terpenes, terpenoids, terpenols). While essential oils remain a rich source of potential antifungals, focus should shift to prioritizing activity from novel compounds outside the commonalities reported here, instead of simply identifying antifungal activity. Further, capitalizing on bigger data approaches can provide significant returns in expediting the identification of active components.

## 1. Introduction

Worldwide, invasive fungal infections are responsible for greater than 1.5 million deaths annually [1,2,3]. *Aspergillus*, *Candida*, and *Cryptococcus* are responsible for a majority of these infections. Invasive aspergillosis is estimated to infect over 200,000 people per year with a mortality rate between 30% and 95%, invasive candidiasis is estimated to infect over 1 million people per year with a mortality rate between 10% and 75%, and invasive cryptococcosis is estimated to infect over 400,000 people per year with a mortality rate between 20% and 70% [2,3,4,5,6,7,8]. Even more dire, fungal infection and mortality rates are increasing despite current treatment options [9,10,11,12]. The increase is largely attributed to the emergence of drug-resistant fungal strains and an increasing immunocompromised population [13,14,15]. Exacerbating the situation, current antifungals generally require long-term adherence and have significant negative side effects. Thus, the need for new, safe, and effective antifungals is urgent and growing.

Phytochemicals have long been a source of medicinal compounds. This is particularly true for plant essential oils, largely due to their abundance in nature, unique composition, and molecular complexity [13,16,17]. Additionally, the location, time of year, environmental exposure, contact with fertilizer or pesticides, and a variety of other factors may significantly impact the chemical composition of plant essential oils [18,19,20,21]. Since plants can gain important benefits, like mitigating development of resistance, from multiple compounds with multiple targets and multiple modes of action (cocktail approach), there is the potential for multiple antimicrobial compounds in an essential oil. Regardless, one of the major bottlenecks facing natural product drug discovery is the identification of active components. Traditionally, this involves bioactive fractionation which requires a significant amount of natural product sample and is resource and time intensive—especially, when a large number of essential oils are involved.

To determine if bigger data methods could increase throughput and reduce the current bottleneck, we expanded a typical screen to include a large number of essential oils while still coupling quantification of antifungal efficacy with molecular characterization. Overall, we tested 82 plant essential oils for antifungal activity against *A. niger*, *C. albicans*, and *C. neoformans*. Minimum inhibitory concentrations (MICs) were determined and GC-MS spectra were acquired for each essential oil. It was found that all of the pathogenic fungi were inhibited to varying degrees. Notably, the susceptibility of *C. neoformans* was extremely high, with 32 of the essential oils tested having an MIC value of 160 ppm or lower. Conversely, seven essential oils showed activity at or below 160 ppm for *A. niger*, while *C. albicans* only had two essential oils showing the same level of inhibitory activity. Comparing the major constituents of the essential oils that showed antifungal activity, the constituents were largely accounted for by similar classes of molecules (i.e., terpenes, terpenoids, terpenols). While more data would be needed to apply more rigorous mathematical approaches (like singular value decomposition or principle component analysis) to deconvolute trends in inhibitory properties, simple exclusionary principles can be implemented to reduce the number of potential active components. Overall, from a bigger data perspective, the priority for essential oil antifungal discovery should focus on compounds outside of the commonalities reported here. Also, bigger data approaches can contribute to more efficient identification of active components if data are available, potentially reducing one of the major bottlenecks faced by traditional natural product drug discovery.

## 2. Results

Essential oils were randomly chosen for inclusion in the study. Oils from a variety of plant families (citrus, spices, evergreens, deciduous tree, shrubs, ornamental) and geographical locations were included. For antifungal activity, we focused on the three pathogens that are responsible for a majority of human fungal infections, namely, those fungi are the filamentous, mold-like ascomycete *Aspergillus niger*, the biofilm forming, yeast-like ascomycete *Candida albicans*, and the encapsulated basidiomycete *Cryptococcus neoformans*.

### 2.1. Fungal Susceptibility

For each of the three pathogenic fungal species studied, MIC values for all 82 essential oils are reported (see Appendix B). Given the unique natures of the fungal pathogens, it is not surprising that significant differences in susceptibility are observed. Most notable is the high degree to which *C. neoformans* is susceptible to inhibition by essential oils, with 37 of 82, or 45%, having MIC values less than or equal to 160 ppm (see Figure 1). In contrast, for *A. niger*, only seven essential oils and for *C. albicans*, only two essential oils show similar levels of inhibition (see Table 1). Using the previously suggested MIC of 100 µg/mL (equivalent to 100 ppm for essential oils) as a cutoff for significant antimicrobial potential to continue the study of crude natural products [22], 15 of the essential oils active against *C. neoformans* are of significant interest. In contrast, by this criteria, *A. niger* was the least susceptible and not significantly inhibited whereas for *C. albicans*, only one essential oil meets the significance criteria.

When considering the range of inhibition, an initial inspection would suggest that narrow-spectrum inhibition is mostly observed. For the 15 essential oils that significantly inhibited *C. neoformans*, only one shows significant activity against *C. albicans* and none against *A. niger*. However, the most potent essential oil that inhibits *C. neoformans* is also the most potent inhibitor of *C. albicans*. Relaxing the significance criteria reveals more commonalities, blurring the narrow spectrum distinction (see Table 2). This is readily apparent from five of the seven essential oils having 160 ppm MICs against *A. niger*, just outside the 100 ppm significance cutoff, inhibiting *C. neoformans* with an MIC at or below 80 ppm. All seven inhibit *C. neoformans* at or below the same 160 ppm level that they inhibit *A. niger*. Therefore, the initial antifungal specificity of essential oils is not as pronounced as it first appears.

### 2.2. Essential Oil Composition

The composition of the essential oils was characterized by GC-MS and the data are shown in Appendix A. From the 82 essential oils, it was found that approximately 750 different compounds could be confidently identified, accounting for greater than 99% of the observed constituents. Focusing on the major constituents, defined as a component being present at 5% or greater of the total composition, we found 157 different major constituents from the 82 essential oils. Of the 157 major constituents, 114 were unique, found in only one of the essential oils (see Figure 2). Therefore, each essential oil, on average, introduced 1.35 new major constituents into the analysis. Of the 43 non-unique major constituents, six were very common, appearing as major constituents in 10 or more of the essential oils (right half of Figure 2). All are terpenes/terpenoids, representatives from one of the major chemical classes common in essential oils [23]. Overall, while there is a considerable degree of commonality in essential oils, there is still a significant degree of difference in terms of major constituents. Thus, foregoing the antifungal activity of common, well known major constituents, essential oils represent a rich source of small molecules for antimicrobial drug discovery.

### 2.3. Major Constituents of Antifungal Essential Oils

Using bigger data approaches to analyze antifungal activity of essential oils was constrained by the uniqueness of the major constituents which generated a large, but sparse data set. Approaches using singular-value decomposition or principle component analysis to correlate major constituents to MIC efficacy require more than one entry for each constituent for meaningful convergence. This can be somewhat overcome using a hierarchical clustering analysis [18] where major constituents are grouped based on chemical similarities and then the groups correlated to antifungal efficacy. While this approach is useful, unique identification of active constituents is still not possible. Given this limitation, we used an exclusionary principle to readily eliminate major constituents that could not be responsible for the antifungal activity. Simply by removing major constituents that have higher concentrations but less potent MIC values than in other essential oils, the number of possible antifungal constituents could be significantly reduced. For the 15 essential oils that significantly inhibited *C. neoformans*, there were a combined total of 48 major constituents. Using the simple exclusion principle, 13 of these major components could be eliminated from consideration, a 27% reduction. For the two essential oils with significant activity against *C. albicans*, there were six major constituents in total. The simple exclusion removed two. Therefore, by considering larger data (i.e., having more data sets) and consistent (at least relatively) MIC and GC-MS measurements, the identification of active components can be expedited. With more data, the convergence of singular-value decomposition or principal component analysis approaches would improve, potentially providing unique identification of antifungal constituents. However, the number of data sets needed for unique identification could not be realistically estimated beyond a rudimentary approximation of several thousand.

### 2.4. Chemical Similarity of Major Constituents

While individual active compounds could not be discerned with complete confidence, structural analysis of the potentially active constituents does provide additional insight. Structures of major constituents found in essential oils demonstrating antifungal activity are shown in Appendix A. Using the clustering capability of ChemMineR [24], the structures of major constituents for essential oils with significant antifungal activity were grouped based on similarity. It should be noted that using the cutoff of 100 ppm meant that the significant oils found for *C. neoformans* accounted for all antifungal activity (since none were found for *A. niger* and the one essential oil for *C. albicans* was also found for *C. neoformans*). Thus, the analysis could be considered general, but really most relevant for and centered on *C. neoformans*.

The chemical similarity clustering revealed different categories, the members of which are henceforth referred to as Bins with a numbering distinction. Four Bins have multiple compounds. Bin 1 consists of sesquiterpenoids, sesquiterpenes, and sesquiterpenols. Bin 3 contains three different decane derivatives. Bin 25 has monoterpenes and monoterpenoids. Bin 20 contains two santalol derivatives. The compounds in their own individual Bins were Bin 13, an aromatic ketone, Bin 15, a monoterpene, Bin 18, a terpenoid, and Bin 27, a monoterpene. Most of these compounds have already been reported in the literature to be major components of multiple essential oils [25,26,27]. Also, numerous reports of antimicrobial activity have been reported for essential oils containing many of these individual constituents. However, no MIC quantification is available for individual constituents against *A. niger*, *C. albicans*, nor *C. neoformans* using the same methods reported here, further emphasizing the need for methodology to more efficiently and effectively identify the active constituents, not just general antifungal activity.

### 2.5. Cedrol: A Test Case for Individual Component Screening

The essential oil from *Cedrus atlantica (2)*, #16 in our study, most potently inhibited both *C. albicans* (80 ppm) and *C. neoformans* (20 ppm). The major constituent cedrol has been reported in other essential oils to have antifungal properties [28,29] but has not been studied in isolation. Therefore, the individual compound was screened against both *C. albicans* and *C. neoformans*. Cedrol was obtained commercially, and samples were prepared in DMSO in both 1% concentrations, exactly as the full essential oils were, and in 0.23% concentrations, which corresponds to the amount of cedrol in the active essential oil. The screening results revealed that commercial cedrol inhibited *C. neoformans*, but was inactive against *C. albicans*. The 0.23% solution did not account for the full activity of the *Cedrus atlantica (2)* essential oil, having an MIC of 160 ppm (compared to the 20 ppm for the essential oil). Thus, it appears that other constituents contribute to the antifungal activity of the entire essential oil. As before, this cocktail approach can provide the advantage of multiple modes of action which limits the development of resistance. Having multiple active components needs to be accounted for in further algorithm development. This example also illustrates the need for antimicrobial susceptibility data to advance essential oil contribution to antimicrobial drug discovery.

## 3. Discussion

The trend towards larger data sets and bigger data methodology is growing in all fields of science. Therefore, application of these methods to essential oils antifungal discovery is natural. Before further discussion of the results, a discussion of the limitations is warranted. To maximize the beneficial outcome, the same laboratory equipment and personnel conducted the GC-MS characterization and antifungal studies under as reproducible conditions as possible. Thus, the 82 essential oil data set used is internally consistent and approaches an ideal case. Difficulties arise when comparing GC-MS and fungal susceptibility data from different studies due to many factors. These include different extraction methods, varying climates that can affect essential oil composition, and different methods to obtain susceptibility results (such as different media, assay techniques, and cell concentrations) [30,31,32,33]. Therefore, caution must be taken when comparing essential oil bioactivity results across various studies, especially when considering interlaboratory differences [34]. Another limitation is that inhibition of only one strain of each pathogenic fungi is reported, a particularly important point for *Cryptococcus neoformans* where only one serotype is represented. Other studies show variations in essential oil antifungal activity exist across different strains of the same fungal species [35,36,37], again requiring further study for meaningful generalization. Even with these caveats, the benefits of essential oils in antifungal drug discovery and use of larger data approaches is readily apparent.

The finding that *C. neoformans* is highly susceptible to essential oil inhibition was encouraging, albeit unrelated to the methodology. While this may have some relation to the capsule and/or fungal membrane, further discussion is beyond the scope of this study. With respect to essential oil composition, comparison to previously published data underscores the need for consistent reporting. For *Cedrus atlantica (2)*, the three individual components found with the highest percentage were alpha-cedrene (31.4%), cedrol (23.52%), and cis-thujopsene (20.45%). From other studies, one reported that the three individual components found with the highest percentage were alpha-pinene (14.85%), himachalane (10.14%), and beta-himachalene (9.89%) [38], a different study reported E-gamma-atlantone (19.73%), E-alpha-atlantone (16.86%), and 5-isocedranol (11.68%) [39], while another reported limonene (29.18%), myrcene (16.9%), and ocimene + alpha-pinene (18.6%) [40]. For all studies, the essential oil was collected in or around Morocco via hydrodistillation. Similarly, previous studies have reported *Coriandrum sativum* to have activity against *C. albicans* [41,42], which was not observed herein. However, antifungal activity of *C. sativum* essential oil was noted against *C. neoformans*. Therefore, antifungal activity must be associated with the major constituents, not the essential oil.

Our investigation of bigger data approaches was constrained by the diversity of major constituents in the essential oils tested. In many cases, a major constituent of an essential oil with potent antifungal activity was only present in that oil. Thus, while the potential exists, discrimination of unique antifungal activity in terms of individual major constituents is not feasible without more data. While underscoring the utility of essential oils as being a large source of chemically different compounds, we and others [28,43,44] were limited by active compound identification. Even limited to a simple exclusion, using bigger data tools to facilitate analysis of essential oil data had an immediate positive impact.

With regard to identification of active compounds, it is necessary to bring up the possibility that more than one component of the essential oil may be responsible for the antifungal activity. Generally, it is the major component(s) that will be responsible for bioactivity, and not those found in trace amounts [45]; therefore, our focus has been centered on major constituents. Previous studies report that one or two of the major components found in an essential oil will be responsible for the bioactivity seen against a microorganism [21,23]. However, it has been shown in other studies that a synergistic effect between several different essential oil components is responsible for the full bioactivity [46,47]. It has also been shown across several different studies that some essential oils possess a single mechanism of action, while others possess two or more [45], meaning that sometimes interactions amongst different constituents are important for biological activity. As shown for cedrol, it is possible that even after identifying the potentially active major component(s), full activity may not be found and synergy between multiple essential oil components must be considered. Thus, bigger data approaches and algorithms must have the ability to account for such combinatorial outcomes.

## 4. Materials and Methods 

### 4.1. Essential Oils

Essential oils were obtained commercially (doTerra, Pleasant Grove, UT, USA). All oils were tested as 1% solutions in DMSO made by suspending 10 µL of essential oil in 990 µL of DMSO. 

### 4.2. Fungal Cultures

Pathogenic fungal strains of *Aspergillus niger* (ATCC #16888), *Candida albicans* (ATCC #18804), and *Cryptococcus neoformans* 24,067 (serotype D or var. *neoformans*) were utilized. *A. niger* cultures were initially grown on potato dextrose agar (PDA) at room temperature (~22 °C) before the conidia were filtered and resuspended in potato dextrose broth (PDB). Conidia density in the broth was adjusted with PDB to a density of 4000 conidia/mL from previously reported methods [48]. *C. albicans* and *C. neoformans* were grown on potato dextrose agar plates for 48–72 h, respectively, before a single colony was picked and grown in PDB to create initial cultures. Cells from these cultures were diluted to a concentration of 4000 cells/mL using RPMI 1640 (Roswell Park Memorial Institute) buffered with 167 mM MOPS (3-(N-morpholino)propanesulfonic acid) at pH 7.0. 

### 4.3. Microdilution

Screening was performed according to CLSI guidelines. For all fungi, the 96-well microdilution method was used with a final well volume of 200 µL. For *C. albicans* and *C. neoformans*, 100 µL of RPMI was first added to all wells. Subsequently, 100 µL of the antifungal sample or control was added to the respective well in row A, mixed and then serially diluted in each row of the plate. To this, 100 µL of the initial fungal inoculum (described above) was added, for a final cell concentration of 2000 cells/mL. The microplates for *C. albicans* and *C. neoformans* were incubated at 37 °C for 24 and 72 h, respectively, before being analyzed. For *A. niger*, 100 µL of RPMI was added to all wells followed by serial 2-fold dilution of test samples or controls. Added last was 100 µL of the *A. niger* conidia solution (described above), for a final conidia concentration of 2000 conidia/mL. The *A. niger* microplates were incubated at room temperature for 7 days before being analyzed. All MICs were determined from triplicate measurements. RPMI was used as a negative control while 100% DMSO and 25 µg/mL amphotericin B were used as positive controls. 

### 4.4. GC-MS

The essential oils were analyzed by GC-MS using a Shimadzu GCMS-QP2010 Ultra operated in the electron impact (EI) mode (electron energy = 70 eV), scan range = 40–400 atomic mass units, scan rate = 3.0 scans/s, and GC-MS solution software. The GC column was a ZB-5 fused silica capillary column with a (5% phenyl)-polymethylsiloxane stationary phase and a film thickness of 0.25 μm. The carrier gas was helium with a column head pressure of 552 kPa and flow rate of 1.37 mL/min. The injector temperature was 250 °C and the ion source temperature was 200 °C. The GC oven temperature was initially 50 °C followed by the temperature increased at a rate of 2 °C/min to 260 °C. A 5% *w*/*v* solution of the sample in CH_2_Cl_2_ was prepared and 0.1 μL was injected with a splitting mode (30:1). Identification of the oil components was based on their retention indices determined by reference to a homologous series of *n*-alkanes, and by comparison of their mass spectral fragmentation patterns with those reported in the literature [49] and stored in our in-house MS library.

### 4.5. ChemMineR

PubChem IDs for the potential active antifungal major constituents were uploaded to the ChemMineR server. Binning was performed using the default similarity cutoff of 0.4. The reported binning output was converted to a graphical figure manually.

### 4.6. Cedrol from Commercial Sources

Cedrol (TCI America, Portland, OR, USA) was obtained commercially. For a 1% solution, 10 µL of cedrol was added to 990 µL of DMSO. For the 0.23% solution, 23 µL of the 1% solution was added to 77 µL of DMSO.

## 5. Conclusions

With the need for new treatment options for fungal infections, essential oils offer a promising avenue for antifungal discovery and development. A growing essential oil knowledge base enables efforts to reduce the bottleneck of active component identification. Our investigation of bigger data approaches underscores this utility, showing that correlating efficacy to composition is possible to facilitate identification of active constituents. Currently a complement to, not substitute for, bioactive fractionation, this methodology holds great potential to expedite the identification of active constituents. As more and larger data sets become available, bigger data approaches will improve. (A curated database with relatively consistent data would greatly improve the performance of any such big/bigger data approaches.) The high susceptibility of *C. neoformans* to essential oil inhibition was also encouraging, serving to emphasize the priority for antifungal essential oil screening should move towards oils with compounds outside the common, generally active classes reported here. Overall, while essential oils remain a staple source for antimicrobial inhibitors, improving processes that efficiently identify lead candidates is of great benefit, expediting the next generation of drug discovery.

## Figures and Tables

**Figure 1 molecules-24-02868-f001:**
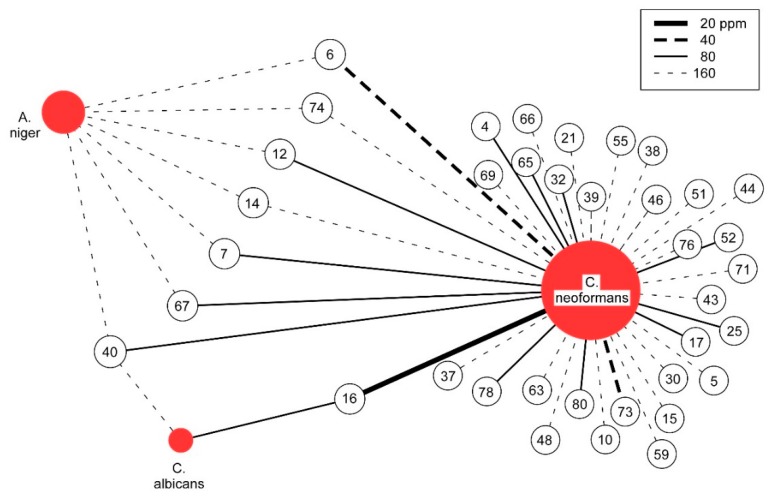
Graphical Analysis of Antifungal Activity. The antifungal activity of essential oils with MIC values at or below 160 ppm are depicted. Unshaded circles represent essential oils, numbered as noted in Appendix B. The strength of MIC is indicated by the line (see legend).

**Figure 2 molecules-24-02868-f002:**
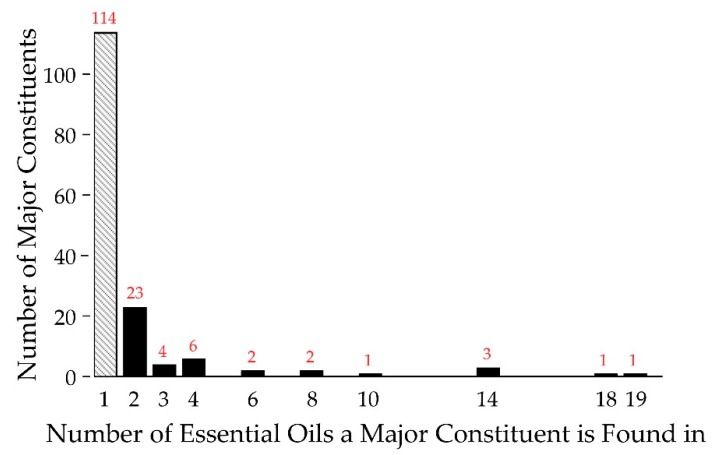
Major Constituent Frequency Histogram. Shown are the frequencies of occurrence for the major constituents of essential oils. A total of 114 major constituents are unique to one essential oil (far left, in gray), while one major constituent is found in 19 essential oils (far right). Red numbers indicate the population of a given column.

**Table 1 molecules-24-02868-t001:** Fungal Susceptibility to Essential Oils. The number of essential oils having the indicated MIC value are shown for each fungal pathogen. Those meeting the MIC < 100 ppm significance criteria are shaded.

Fungal Pathogen	MIC = 160 ppm	80 ppm	40 ppm	20 ppm
*A. niger*	7	0	0	0
*C. albicans*	1	1	0	0
*C. neoformans*	22	12	2	1

**Table 2 molecules-24-02868-t002:** Essential Oil Efficacy against Pathogenic Fungi. The more stringent significance cutoff of 100 ppm applies to *C. neoformans*; the asterisk (*) indicates a more relaxed susceptibility cutoff of 160 ppm for *C. albicans* and *A. niger*. N/A indicates no activity beyond the vehicle carrier control.

Essential Oil Name	*C. neoformans*	*C. albicans* *	*A. niger* *
*Cedrus atlantica (2)*	20	80	
*Amyris balsamifera*	40	N/A	160
*Santalum spicatum*	40	N/A	
*Ferula galbaniflua*	80	160	160
*Ajowan Trachyspermum ammi*	80		
*Callitris intratropica*	80	N/A	160
*Pogostemon cablin*	80	N/A	160
*Citrus aurantium ssp. Amara*	80	N/A	N/A
*Chamaecyparis obtuse*	80	N/A	N/A
*Aquilaria sinensis*	80	N/A	160
*Kunzea ericoides*	80	N/A	
*Vitex agnuscastus*	80	N/A	N/A
*Citrus clementina*	80		N/A
*Coriandrum sativum*	80	N/A	N/A
*Turnera diffusa*	80	N/A	N/A

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
