# Peer review of "Bigger Data Approach to Analysis of Essential Oils and Their Antifungal Activity against Aspergillus niger, Candida albicans, and Cryptococcus neoformans"

_molecules, 2019, doi:10.3390/molecules24162868_

Round 1

Reviewer 1 Report

Review of the article: Bigger Data Approach to Analysis of Essential Oils and their Antifungal Activity against Aspergillus niger, Candida albicans, and Cryptococcus neoformans

Manuscript ID: molecules-537674

I have read the manuscript with great interest, and I have to admit that I have a big problem with the final decision. All experiments were well (excellent) planned and performed and the obtained results are very interesting. But the authors have not compared/discussed the obtained results with the results presented by other researchers – there are a plenty of publications presenting antifungal activity of essential oils (some of them were presented in MDPI journals including Molecules or Pathogens) which could be used for discussion. In my opinion in the current form (without any discussion of obtained results) the article cannot be accepted. I would propose the authors preparing revised version of the manuscript – with discussion.

Other weaknesses of the manuscript are:

1.      Using only one strain of each of fungal pathogens for investigation (some authors have shown important differences of activity of essential oils against C. albicans, C. glabrata and other fungal pathogens) – you can find some of examples in articles presented in Molecules. Of course I understand that the authors will not be able to repeat all test with new strains, but they should mention in discussion that other authors observed important differences of susceptibility of isolates of the same species against oils isolated from one botanical source.

2.      The authors should mention about some limitations of essential oils – please read the paper presented by Serra and coworkers (doi: 10.3390/pathogens7010015.)

3.      In my opinion identification of cedrol as potential antifungal agent and investigation of activity of this component is an example of excellent planning of research by the authors. But I also suggest that they should mention in (introduction or discussion) that natural products contain many different ingredients and antimicrobial activity is rarely caused by one particular component. Usually it is caused by several ingredients that exhibit different modes of action – this multi-target mode of action is important benefit of natural products compared to antibiotics (which exhibit activity against one molecular target), it protects against development of resistance.

4.      Conclusion should be changed.

Important benefits of the revised manuscript – well planned and performed research and interesting results.

Final opinion – major revision

Reviewer 2 Report

Authors describes the effect of essential oils on pathogenic fungi by bigger data approach. The chemistry of the oil is not well described. Please report GC-MS analysis of the each investigated oil. Abstrat and conclusion section must be rewrite. Please improve discussion section.

Authors describes the Antifungal Activity against Aspergillus 4 niger, Candida albicans, and Cryptococcus neoformans of a series of essential oils. The major concern regarding the incomplete presentation of essential oil chemistry . Authors cite the use of GC-MS analyisis but data are not fully reported. Moreover if 80 essentail oils are investigated why the name is not reported? Maybe data should be inserted as supplementary material.

Lines 124-125 reported that alpha-pinene (appearing in 19 essential oils), limonene (18), 1-8-cineole (14), beta-pinene (14), 125 linalool (14), and sabinene (10) are main constituents of several oil. This information is not new. Moreover it is well know by the literature that monoterpenes are carachterzied by antimicrobial activities.

Fig. 2 is not necessary since no important informations are reported.

Check Table 2 the format presentation is not correct

Line 147 “For C. neoformans, from the 15 essential oils that show significant 147 anticryptococcal activity, there were 48 major constituents.” This sentence is not clear please REWRITE.

Line 182 “Cedrol was obtained commercially” information regarding Cedrol supplier should be reported in material and method section here just “cedrol” is in half.

Lines 186-187. A synergistic action of essential oil should be considered and discussed (please use the literature data.

Rewrite paragraph 3.1 a complete description of essential oils used should be inserted. Are commercial?

Lines 152-154 is not clear please explain better the statistical approach and why it is a useful method.

The abstract is too descriptive please revise it in a more concise way and at the same time in a more scientific way. Lines 12-15 are more suitable for Introduction section.

Improve discussion section since no comparison with literature data are reported. Expecially for paragraph 2.4

Lines 187-189 is not clear explain.

Conclusion section should reported the novelity of the study. Now is too similar to introduction please rewrite it.

Round 2

Reviewer 1 Report

The authors have taken into account all my suggestions and remarks (the presented discussion is quite interesting and well prepared). The manuscript can be accepted.

Reviewer 2 Report

The revised version of the article is now suitable for publication.